# Evaluation of TFR-1 Expression in Feline Mammary Cancer and In Vitro Antitumor Efficacy Study of Doxorubicin-Loaded H-Ferritin Nanocages

**DOI:** 10.3390/cancers13061248

**Published:** 2021-03-12

**Authors:** Nicolò Rensi, Alessandro Sammarco, Valentina Moccia, Alessandro Calore, Filippo Torrigiani, Davide Prosperi, Maria Antonietta Rizzuto, Michela Bellini, Raffaella De Maria, Federico Bonsembiante, Silvia Ferro, Rossella Zanetti, Valentina Zappulli, Laura Cavicchioli

**Affiliations:** 1Department of Comparative Biomedicine and Food Science, University of Padua, 35020 Padua, Italy; alessandro.sammarco@unipd.it (A.S.); valentina.moccia@phd.unipd.it (V.M.); alessandro.calore.1@studenti.unipd.it (A.C.); filippo.torrigiani@unipd.it (F.T.); federico.bonsembiante@unipd.it (F.B.); silvia.ferro@unipd.it (S.F.); rossella.zanetti@unipd.it (R.Z.); valentina.zappulli@unipd.it (V.Z.); laura.cavicchioli@unipd.it (L.C.); 2Department of Neurology and Radiology, Massachusetts General Hospital, Harvard Medical School, Boston, MA 02114, USA; 3NanoBiolab, Department of Biotechnology and Bioscience, University of Milano-Bicocca, 20126 Milan, Italy; davide.prosperi@unimib.it (D.P.); maria.rizzuto@unimib.it (M.A.R.); michela.bellini@unimib.it (M.B.); 4Department of Veterinary Science, University of Turin, 10095 Turin, Italy; raffaella.demaria@unito.it; 5Department of Animal Medicine, Productions and Health, University of Padua, 35020 Padua, Italy

**Keywords:** TFR-1, feline, mammary gland, carcinoma, metastasis, doxorubicin, H-Ferritin, nanocage

## Abstract

**Simple Summary:**

Transferrin receptor one (TFR-1), recognized by ferritin, is overexpressed in many tumor cells. This feature has been exploited to produce a selective overload of drugs within tumor cells by creating an engineered ferritin nanocage loaded with doxorubicin (HFn(DOX)). This bionanotechnology has been tested in human cancer, but there are no studies in veterinary oncology. This work, after evaluating the expression of TFR-1 in feline tumors, demonstrated for the first time the effectiveness in vitro of this nanocage in animals. These results confirm that engineered bionanocages also offer unprecedented opportunities for animal cancer to be applied in veterinary medicine and in comparative studies including spontaneous animal models of cancer.

**Abstract:**

The transferrin receptor 1 (TFR-1) has been found overexpressed in a broad range of solid tumors in humans and is, therefore, attracting great interest in clinical oncology for innovative targeted therapies, including nanomedicine. TFR-1 is recognized by H-Ferritin (HFn) and has been exploited to allow selective binding and drug internalization, applying an HFn nanocage loaded with doxorubicin (HFn(DOX)). In veterinary medicine, the role of TFR-1 in animal cancers remains poorly explored, and no attempts to use TFR-1 as a target for drug delivery have been conducted so far. In this study, we determined the TFR-1 expression both in feline mammary carcinomas during tumor progression, as compared to healthy tissue, and, in vitro, in a feline metastatic mammary cancer cell line. The efficacy of HFn(DOX) was compared to treatment with conventional doxorubicin in feline mammary cancer cells. Our results highlighted an increased TFR-1 expression associated with tumor metastatic progression, indicating a more aggressive behavior. Furthermore, it was demonstrated that the use of HFn(DOX) resulted in less proliferation of cells and increased apoptosis when compared to the drug alone. The results of this preliminary study suggest that the use of engineered bionanocages also offers unprecedented opportunities for selective targeted chemotherapy of solid tumors in veterinary medicine.

## 1. Introduction

Feline mammary carcinomas (FMCs) are the third most common neoplasia in intact female cats, after lymphohematopoietic and cutaneous tumors [1]. FMCs share some characteristics with human breast cancer including molecular subtype and several biomarkers, which are used for diagnosis [2,3]. Frequently, FMCs are simple carcinomas, highly aggressive and infiltrative, showing a rapid progression with an interval time between diagnosis and death ranging from 6 to 12 months [4,5,6]. The size of the tumor is one of the most important prognostic factors; thus, cats bearing larger carcinomas have a worse prognosis than cats with smaller nodules [7]. In addition, the tumor size frequently influences the choice of treatment. Traditionally, a complete surgical approach is suggested for smaller FMCs [8], whilst, for cats with nodules larger than 2 cm, a combination of surgery and chemotherapy (cyclophosphamide and doxorubicin (DOX)) is recommended [9]. Nevertheless, the use of DOX is widely debated due to some of its side-effects, such as its cardiotoxicity at high dosages and nephrotoxicity [10,11,12]. Moreover, in humans and animals, DOX is internalized within cells by simple diffusion, which is not a selective transport mechanism. As a result, DOX is assimilated by both healthy and tumor cells with inevitable harmful effects on normal tissues [13]. Therefore, currently, one of the main goals of cancer research is to find selective treatments that could possibly spare healthy cells. In addition, DOX, as with most cytotoxic drugs, is known to frequently induce chemoresistance after the first set of chemotherapy cycles [14]. In this light, in the last few decades, nanotechnology has gained importance, whereby engineered nano-atomic-scale materials are applied in the biomedical field [15]. Some nanoparticles, such as liposomes, have been designed in order to reduce cardiotoxicity effects and deliver molecules and drugs more efficiently and selectively to cancer cells [16]. Indeed, liposomal DOX has been approved and has entered the clinical practice [17].

Ferritin is a ubiquitous protein, physiologically related to iron metabolism and made up of 24 (12 heavy chain + 12 light chain) subunits self-organized in a 12 nm spherical shape with an 8 nm internal cavity [18]. In normal conditions, within its cavity, ferritin can store up to 4500 ions of iron, avoiding cytoplasmic iron overload. Mechanistically, the heavy chain subunit of ferritin is able to cross-bind to the transferrin receptor 1 (TFR-1), a transmembrane glycoprotein of 180 kDa located on every cell, to internalize iron ions [19]. Recently, some researchers designed and applied an engineered human apoferritin nanocage consisting of solely heavy-chain subunits (HFn) to exploit both the nanocage feature that allows compounds to be loaded into its cavity and the specific binding to TFR-1 that triggers an endosomal uptake as a portal of entry into cells presenting this transmembrane receptor [20]. TFR-1 expression is related to iron cell request. Considering that tumor cells grow more rapidly than healthy cells, thus having an increased demand of oxygen and nutrients and, as a consequence, of iron, they also increase TFR-1 expression at both the gene and the protein levels [21]. Indeed, in human oncology, TFR-1 has been identified as overexpressed in many types of tumors such as lung, liver, colon, brain, and breast cancer [22]. Specifically, in human breast cancer (HBC), TFR-1 not only was found overexpressed in tumor tissue when compared to the healthy breast, but its expression also increased with tumor malignancy [23]. The HFn nanocage loaded with DOX was demonstrated to act like a “Trojan horse” to drive the drug into the nucleus of human cancer cells, overcoming the onset of chemoresistance and strongly improving the antitumor efficacy of DOX in a mouse model of metastatic breast cancer [13,24].

In veterinary medicine, the expression of TFR-1 has been poorly considered for tumor targeting. The expression of this receptor was identified in canine lymphoma, where it was found to be less expressed in low-grade T-cell lymphoma compared to high-grade T- and B-cell lymphoma [25]. Focusing on canine and feline mammary gland cancer, TFR-1 was investigated by Marques et al. in 2017, and its presence was identified both in benign and in malignant tumors, but a difference between the two types was not observed [26].

The aims of this study were, firstly, to investigate TFR-1 expression in formalin-fixed paraffin-embedded (FFPE) feline mammary carcinomas (FMCs) and the metastatic lymph nodes. Secondly, TFR-1 gene and protein expression was also evaluated in a feline metastatic mammary cancer cell line (FMCm). Thirdly, this study aimed to test the efficacy of an HFn loaded with DOX (HFn(DOX)) on FMCm as a possible new therapeutic horizon for this often lethal feline malignancy, as well as to expand the realm of nanobiotechnology in veterinary medicine.

## 2. Results

TFR-1 protein and gene expression was identified both in healthy and in tumor mammary gland tissues, as well as in the FMCm cell line.

### 2.1. Immunohistochemistry (IHC)

The IHC analysis revealed both cytoplasmatic and membrane localization of TFR-1 in all mammary gland tissues. A different heterogeneous staining intensity was found between and within samples, indicating different levels of expression (Figure 1). The H-score showed an increased protein level of TFR-1 in feline mammary carcinomas with nodal metastasis (mean 112.28 ± SD 40.51) versus healthy mammary gland tissues (mean 40.07 ± SD 38.95) (*p* < 0.05) (Figure 2). Moreover, the lymph node metastases had a higher level of expression (mean 99.48 ± SD 27.95) than the healthy mammary gland tissues (mean 40.07 ± SD 38.95) (*p* < 0.05). In addition, a trend of an increased TFR-1 expression from healthy mammary glands (mean 40.07 ± SD 38.95) to feline mammary carcinomas without lymph node metastasis (mean 78.72 ± SD 29.41) and to feline mammary carcinomas with lymph node metastasis (mean 112.28 ± SD 40.51) was also evident.

### 2.2. Western Blotting Results

The Western blotting assay (Figure 3) carried out on proteins extracted from FMCm showed a 90 kDa protein, corresponding to the expected TFR-1 molecular weight [27] (Figure 3A). The beta-actin housekeeping gene was evident as a 43 kDa protein in FMCm, MCF-7, and MDA-MB231 cell lines (Figure 3B). The original images of Western blotting can be found in the Appendix A.

### 2.3. Immunofluorescence

The immunofluorescence assay highlighted the presence of TFR-1 on FMCm. The expression of TFR-1 was evident both on the cell membrane and within the cytoplasm (Figure 4).

### 2.4. Flow Cytometry Results

The morphology scatter plot (Figure 5A) highlighted that all the events had a cloud-shaped distribution, typical of cells with variable size and low complexity. Therefore, it was evident that all events belonged to the same cell type (Figure 5A). Moreover, the expression of TFR-1 was identified in 95% of the cells (Figure 5B). The negative control (Figure 5C) confirmed antibody specificity.

### 2.5. RT-PCR and Sequencing

The RT-PCR on FMCm revealed a messenger RNA (mRNA) fragment amplification of the expected size (526 bp) (Figure 6). The sequencing showed that the fragment amplified had 99% homology with the *Felis catus* transferrin receptor gene (*TFRC*) (sequence identifier (ID): NM_001009312.1) and 80% homology with the *Homo sapiens* transferrin receptor (TFRC) (sequence ID: NM_001128148.3).

### 2.6. Cell Proliferation Assay

In order to investigate the therapeutic efficacy of HFn(DOX), a proliferation assay was performed. Interestingly, the proliferation rate of FMCm treated with HFn(DOX) was lower (*p* < 0.05) than that treated with DOX at 0.01 μM, 72 h after treatment (Figure 7A) and at 0.1 μM, 48 h and 72 h after treatment (Figure 7B), suggesting that the therapeutic efficacy of HFn(DOX) is higher than that of DOX at low–medium concentrations. Conversely, no differences between cells treated with HFn(DOX) and DOX were found when using a higher concentration (1 μM) (Figure 7C). Moreover, FMCm treated with HFn showed the same proliferation rate as the control Roswell Park Memorial Institute (RPMI), suggesting that the effect of HFn(DOX) on proliferation was not due to the nanocage itself.

### 2.7. Cell Apoptosis Assay

In order to determine if the proliferation arrest was related to cell death, the rate of apoptosis was measured. The results showed that HFn(DOX) was more effective than DOX alone in inducing apoptotic cell death at 0.1 and 0.01 μM 72 h after treatment (Figure 8).

## 3. Discussion

In oncology research, one of the main goals is to find selective therapies able to treat patients while avoiding or minimizing detrimental effects on healthy cells [28]. To achieve this, numerous studies have addressed the investigation of selectively overexpressed molecules to target diseased cells [16,29]. Among these molecules, few studies have investigated overexpression in cancer cells of specific receptors, including TFR-1, to exploit them using nanotechnology to selectively deliver anticancer drugs into tumor cells [11,30].

It is known that tumor cells have a higher metabolic activity compared to healthy cells; hence, they have a higher demand for nutrients and iron, which leads to an increase in the TFR-1 expression to maintain their homeostasis during the rapid growth rate [31]. Similarly to HBC, we observed that the TFR-1 protein expression in feline mammary gland tissues was significantly enhanced in the lymph node metastases and in the tumors that developed lymph node metastases as compared to healthy tissues and carcinomas without lymph node metastasis [23]. Even though this study was performed on a limited number of samples, these results seem to indicate that there is also an increased protein expression of TFR-1 in cats along with mammary tumor progression, consistent with HBC behavior [32]. These IHC findings, on the other hand, were slightly different from another study on mammary neoplasms [26], in which the authors did not find any difference in TFR-1 protein expression between malignant and benign mammary gland tumors in cats and dogs. However, in that study, benign tumors were compared with the malignant ones, but carcinomas of different histological grades were not compared and neither were metastatic versus nonmetastatic malignancies, while healthy tissue was not included in the study. In our study, benign mammary tumors were not included since they are extremely rare in cats, but we compared different malignant tumor groups, including lymph node with metastases, and healthy tissues to investigate differences during malignant exacerbation and the metastatic progression. Our results were similar to those found in canine lymphoma where high-grade malignancies expressed a higher TFR-1 protein level than lower-grade lymphoma [23,25,26]. In our case, no specific association was found among tumor histological grade, size, or lymphovascular invasion (data not included), possibly because of the limited number of samples; nevertheless, there was an association with the metastatic tumor progression indicative of a more malignant behavior.

The TFR-1 overexpression in malignant tumors has already been exploited in human medicine to internalize anticancer drugs more selectively and at a higher dosage in tumor cells by applying nanotechnology vectors, such as HFn [19,33]. It is worth noting that a first set of nanoparticle-based chemotherapeutics developed in advanced-stage clinical trials were designed to address TFR-1 as a preferential molecular target for cancer treatments [34,35]. Consistently, our results confirmed a higher expression in malignancies and metastatic lesions in feline mammary tissues. Thus, to also explore the potential of HFn(DOX) as an innovative targeted nanodrug for the treatment of aggressive/metastatic breast cancer in pets, we moved forward to investigate the interaction of HFn with TFR-1 and the antitumor effect of HFn(DOX) in feline cell cultures.

We first demonstrated the TFR-1 expression on FMCm by both flow cytometry (FC) and immunofluorescence (IF). It is known that TFR-1 is expressed on breast cancer cell lines that we included as controls within the Western blot. We could confirm that the receptor is also present on the feline mammary cancer cell line, apparently with a similar level of expression to that seen in MCF-7 and MDA-MB-23 cells. It is known that TFR-1 is a transmembrane glycoprotein with three different domains: external, transmembrane, and internal [36]. Moreover, the physiological pathway of TFR-1 internalization provides that, after iron binding, the TFR-1/Fe complex is internalized into the cytoplasm via endosome formation, in which the TFR-1 intracellular domain is oriented facing the cytoplasm [20,37]. Since the antibody binds to the internal domain of TFR-1 after cell permeabilization, the expected signals were correctly visualized at IF both close to the membrane and in the cytoplasm.

We next focused on the treatment of FMCm to test the efficacy of exploiting TFR-1 expression to obtain an intracellular drug overload [38]. In this study, cells treated with HFn(DOX) had lower absorbance intensities compared to cells treated with DOX alone, when using a low–medium drug concentration. The absorbance measurements were used to the quantify the formazan products formed by the dehydrogenase enzymes in metabolically active cells, as the absorbance is directly proportional to the number of living cells [39]. Therefore, these results indicated that HFn(DOX) was able to induce a higher antiproliferative effect compared to DOX alone. To demonstrate that these results were not merely due to a blockage of TFR-1 activity, a control experiment using unloaded HFn was conducted, confirming that no significant decrease in cell proliferation could be attributed to the HFn nanocage. In human medicine, it has been shown that cancer cells rapidly internalize the nanoparticle via TFR-1 binding using endocytosis promoted by clathrin-coated pit intake, which normally happens during the physiological iron intake [13,40]. In our study, the higher antiproliferative effect induced by HFn(DOX) compared to the administration of DOX alone was evidenced at 48 h and 72 h post treatment at 0.1 µM DOX concentration. When considering the treatment time plan, the doubling time (DT) of cells under investigation is relevant. FMCm cells have a DT of 29 h [41]; therefore, a 48 h period was needed to show the antiproliferative effect. The concentration of the drug is also important. It was noted that, at concentrations as low as 0.01 µM, the antiproliferative effect of HFn(DOX) was highlighted only after 72 h of treatment, probably because of the time needed to accumulate enough drug. In contrast, no difference in cell proliferation was noted between HFn(DOX) and DOX at 1 µM. It is plausible that, at this concentration, the cells had an excessive accumulation of DOX in both treatments, causing a similar extent of cell death [13]. Interestingly, the effect of HFn(DOX) on FMCm at 0.1 µM dosage at 48 h was similar to that seen with HBC cell lines [13]. 

It could be assumed that cells treated with HFn(DOX) are less alive than cells treated with DOX alone. The additional assay on apoptosis confirmed that the apoptotic pathway is more active when applying HFn, compared to only DOX. This difference was identified only 72 h after treatment at 0.1 µM and 0.01 µM. Instead, at 48h after treatment, the difference was not evident. This might be due to the fact that HFn(DOX) can either induce other cell death pathways or simply reduce the metabolic activity and proliferation of the cells with no yet measurable evidence of apoptosis after a shorter time. In this study, necrosis was not evaluated since the main mechanism of action of doxorubicin is to activate apoptosis by intercalating into DNA and inhibiting the topoisomerase enzyme [42]. Moreover, calculating the rate of necrosis after 48 h and 72 h may not be reliable since a process defined as “late apoptosis/secondary necrosis” has been described to occur in vitro. This process produces membrane disruption of apoptotic bodies that, in cell culture conditions, cannot be removed by phagocytic cells; therefore, necrosis evaluated with propidium iodide can be overestimated [43]. 

According to these preliminary results, HFn appears as a promising nanocarrier to be developed for the delivery of breast cancer chemotherapeutics in cats. Indeed, by binding selectively and specifically to TFR-1, HFn allowed DOX overload and, therefore, a higher cytotoxic effect on tumor cells in comparison to free DOX.

Interestingly, in a mouse model of HBC, it was demonstrated that, upon treatment with HFn(DOX), the drug concentration within the tumor mass was about 10 times higher than after the administration of free DOX with a single dose injection [44]. Moreover, HFn consists of an assembly of ferritin subunits, which are physiologically available within the organism; for this reason, its exogenous administration did not caused any adverse response [19]. These results support the chance to use HFn bionanotechnology to also limit DOX widespread distribution in healthy tissues, thus preventing the drug’s cardiotoxicity and protecting the host from common chemotherapy side-effects [24].

Considering the promising implications in veterinary medicine and our preliminary results on an FMCm cell line, work is ongoing to confirm the selective binding of HFn to TFR-1. Confocal microscopy will give us the opportunity to see the actual binding of the nanocage to TFR-1, as well as allow us to follow the internalization of HFn inside tumor cells. Additionally, an in vivo mouse model with a feline cell line would also allow confirming the hypothesis that the overload of drugs is selective for the tumor and spares the healthy tissue.

## 4. Materials and Methods

### 4.1. Histology and Immunohistochemistry

A total of 27 histological feline mammary gland tissues and lymph nodes were selected from the archive of the Diagnostic Service of Veterinary Pathology of the University of Padua, Italy. Since all the samples were collected as part of routine clinical activity, no ethical committee approval was needed. Samples were obtained by surgical removal. Tissues were fixed in 4% formalin and embedded in paraffin (FFPE), hematoxylin/eosin-stained, and subsequently independently evaluated by two pathologists (L.C., V.Z.). Surgery was performed as a therapeutic treatment and, therefore, no additional sampling or owner consensus was required. The neoplastic lesions were classified into four categories, following the World Health Organization (WHO) classification, recently updated [45]: healthy mammary gland tissue (eight cases), feline mammary carcinoma without lymph node metastasis (seven cases), feline mammary carcinoma with lymph node metastasis (seven cases), and the tributary lymph node that had the metastasis of the primary mammary simple carcinoma (six cases).

For the immunohistochemistry (IHC), sections of 4 µm from the 27 specimens were prepared to analzse and localize TFR-1 protein expression. The staining was performed using a semiautomatic immunostainer (Ventana Benchmark XT, Roche-Diagnostic). All reagents were dispensed automatically except for the primary antibody, which was manually added. A kit containing the secondary antibody and a horseradish peroxidase (HRP)-conjugated polymer which binds mouse primary antibody (UltraView Universal DAB, Ventana Medical Systems, Oro Valley, AZ, USA) was used. An anti-human TFR-1 mouse monoclonal antibody (clone H68.4 Thermo Fisher Scientific catalog number 13-6890) at a 1:25 dilution with 32 min of incubation at 37 °C was used as a primary antibody. The counterstaining with hematoxylin was automatically performed. A semiquantitative immunostaining evaluation was performed with an optical microscope, whereby 10 fields at 40× were randomly chosen and only epithelial cells were counted. Specifically, membranous/cytoplasmatic immunolabeling was measured, considering both the percentage and the staining intensity (0 = no staining, 1 = weak staining, 2 = moderate staining, and 3 = strong staining) of positive cells. Data were then combined to obtain an H-score, which sums the percentage of weakly positive cells, the percentage of moderately positive cells multiplied by 2, and the percentage of strongly positive cells multiplied by 3. In this way, a range score from to 0 to 300 can be obtained [46].

### 4.2. Cell Culture

A feline metastatic mammary cancer cell line (FMCm) was used in this study to show the presence of TFR-1 and to test the efficacy of HFn. The cell line was obtained from the inguinal lymph node metastasis of a feline mammary simple carcinoma [41]. Two breast cancer cell lines (MCF-7, MDA-MB231) obtained from the American Type Culture Collection (ATCC) were included in the study as positive controls. FMCm cells were grown in Gibco^®^ Advanced RPMI 1640 1× (Thermo Fisher Scientific, Waltham, MA, USA). MCF-7 and MDA-MB-231 cells were grown in Gibco^®^ Dulbecco’s Modified Eagle Medium 1× (DMEM, Thermo Scientific, Waltham, MA, USA). In both cell lines, 10% fetal bovine serum (FBS, PAN™ BIOTECH, Aidenbach, Germany) and 1% of penicillin–streptomycin (Corning^®^ 100 mL Penicillin-Streptomycin Solution, 100×, Corning, NY, USA) were supplemented. Cells were incubated at 37 °C in a humidified atmosphere containing 5% CO_2_. Cells, at confluence, were harvested using 0.25% trypsin- etilendiamminotetraacetic acid (EDTA) (Gibco^®^ trypsin-EDTA 1×, Thermo Fisher Scientific) and prepared for the subsequent analysis.

### 4.3. Protein Extraction and Western Blotting Analysis

In order to evaluate the cross-reactivity for the feline species of the anti-human TFR-1 mouse monoclonal antibody, a Western blotting analysis was carried out. Briefly, FMCm cells were lysed with RadioImmunoPrecipitation Assay buffer (RIPA) Thermo Fischer Scientific, Waltham, MA, USA), adding protease inhibitor (Sigma Aldrich, St. Louis, MO, USA). A Bicinchoninc Acid Assay, (Pierce BCA^TM^ protein Assay Kit, Thermo Fischer Scientific, Waltham, MA, USA) was used to calculate the protein concentration, and Western blotting analysis was performed following the protocol previously described by Sammarco et al. in 2018 [47]. Briefly, 20 µg of proteins after denaturation at 95 °C for 5 min were resolved by NuPAGE 4% to 12% Bis-Tris gel (Thermo Fisher Scientific) and were transferred onto a nitrocellulose membrane (Invitrogen, Thermo Fischer Scientific, Waltham, MA, USA). By 1 h at room temperature (RT) in 5% nonfat dry milk in Tris-Buffered saline, Tween 20 (TBS-T) (TBS + 0.05% Tween-20), nonspecific binding sites were blocked. After these steps, blots were incubated at 4 °C overnight with the same mouse monoclonal antibody against TFR-1 used for IHC (clone H68.4 Thermo Fisher Scientific catalog number 13-6890), at a dilution of 1:300. The beta-actin mouse monoclonal antibody (C4) sc-47778 Santa Cruz Biotechnology, Dallas, TX, USA) was used as a housekeeping control. The beta-actin antibody was incubated with blots at 4 °C overnight at a dilution of 1:1000.

After overnight incubation, the membrane, after three washes in TBS-T, was incubated with peroxidase-conjugate secondary antibody 1:3000 (GE Healthcare Life Science, Buckinghamshire, UK) for 1 h at RT. The reactive bands were visualized with iBright 1500 (Thermo Fischer Scientific, Waltham, MA, USA) using a chemiluminescence detection kit (SuperSignal West Pico Chemiluminescent Substrate, Thermo Fischer Scientific, Waltham, MA, USA).

### 4.4. Immunofluorescence

In order to study TFR-1 expression on FMCm, immunofluorescence (IF) was performed. Briefly, 25,000 cells were cultured in their medium on a glass coverslip in a 24-well plate. At approximately 80% confluence, cells were washed quickly with PBS, fixed in 4% paraformaldehyde, and permeabilized with 0.1% Triton X-100 (Sigma Chemical Co., St. Louis, MO, USA). Nonspecific binding sites were blocked with 1% bovine serum albumin (BSA, Sigma-Aldrich, St. Louis, MO, USA). Then, the same mouse monoclonal antibody against TFR-1 used for IHC (clone H68.4 Thermo Fisher Scientific catalog number 13-6890) was added at a 1:250 dilution for 3 h at RT. A secondary goat anti-mouse immunoglobulin G (IgG; heavy (H) + light (L) chain) Superclonal™ antibody (AlexaFluor^®^ 555 conjugated Thermo Fisher Scientific, catalog number A32727) diluted 1:2000 was used for 45 min at RT. The immunolocalization images were taken with a Leica DM 4000B microscope, equipped with a Leica DC300F Camera and Leica Image Manager 50 software (Leica Microsystem, Wetzlar, Germany). The TFR-1 positivity was confirmed by the red colorimetric reaction of the cell membrane/cytoplasm.

### 4.5. Flow Cytometry

Three biological replicates of FMCm were analyzed by flow cytometry (FC) for TFR-1 expression. Cells were cultured in their medium and, when at confluence, 500,000 cells were harvested and resuspended in 500 µL of RPMI 1640 + sodium azide + FBS (1000 cells/µL). For each tube, 50 µL of cell suspension was used. Since TFR-1 is a transmembrane receptor, cells were permeabilized using the eBioscience™ FoxP3/Transcription factor staining buffer set (ThermoFisher scientific, Waltham, MA, USA) following the manufacturer’s instruction. After the permeabilization, cells were incubated for 1 h at 4 °C with the anti-human TFR-1 mouse monoclonal antibody (dilution 1:100; clone H68.4 from Thermo Fisher Scientific catalog number 13-6890). After incubation with the primary antibody, the cells suspensions were washed and incubated with a goat anti-mouse IgG (H + L) (Alexa flour® 647 secondary antibody Thermo Fisher Scientific, catalog number A32733) at a 1:500 dilution for 45 min at 4 °C. Samples were washed and resuspended in 900 µL of PBS for the acquisition. A separate aliquot of cell suspension, in which no secondary antibody was included, was used as a negative control, in order to eliminate nonspecific labeling from the analyses. The data acquired by flow cytometer CyFlow Space (Partec-System, Sysmex Europe GmbH, Norderstedt-Amburgo, Germany) were analyzed with the open-source software FCSalyzer (version 0.9.16-alpha). For each replicate, 20,000 events were acquired. The morphology and the complexity of cells were evaluated in forward scatter (FSC) and side scatter (SSC), while TFR-1-positive cells were identified on fluorescence channel 5 versus FSC.

### 4.6. TFRC Gene Expression and Sanger Sequencing

RNA was extracted from 5 × 10^6^ FMCm cells using an RNeasy Mini kit (Qiagen, Hilden, Germany) following the manufacturer’s instructions. Then, 500 ng of total RNA was reverse-transcribed using the RevertAid First Strand complementary DNA (cDNA) Synthesis kit (Thermo Scientific, Waltham, MA, USA) according to the manufacturer’s instructions. Primers were designed with PRIMER-BLAST software, National Center for Biotechnology Information (NCBI) on an exon–exon junction (*TFRC* F5 forward primer: GGTGCCAGTGTCACAAAACC; *TFRC* F3 reverse primer: ATGCCACATAGCCCTCTGGA). RT-PCR was carried out using a Phire™ Hot Start II DNA Polymerase (Thermo Fisher Scientific) according to the manufacturer’s instructions (Mj research PTC-200 Thermal cycler). The expected PCR product (526 bp) was visualized by 2% agarose gel electrophoresis and was then treated with Exosap-IT PCR product cleanup (Applied Biosystem, Foster City, CA, USA) to allow purification for sequencing. Sanger sequencing was performed by BMR Genomics (University of Padua, Italy).

### 4.7. Engineered HFn, FMCm Cell Treatment, and Cell Proliferation Assay

Previously characterized HFn(DOX) [13,24] endowed with high biocompatibility and low cellular toxicity [18,48] and an unloaded HFn were used as the nanodrug and control, respectively, for this study.

HFn(DOX), DOX alone, HFn alone, and RPMI were used at different concentrations on the FMCm, and, to establish the effect on cell proliferation, a CellTiter 96^®^ AQueous One Solution cell proliferation assay (MTS, Promega, Madison, WI, USA) was performed after treatments. Briefly, 5000 cells were seeded in each well of a 96-well plate. Then, 24 h after seeding, cells were incubated at three different drug concentrations (0.01 µM, 0.1 µM, and 1 µM) considering three time points (24 h, 48 h, and 72 h). A cell proliferation assay was performed to test treatment outcomes on FMCm. After each time point, 20 µL of MTS was added to the cells, and the plate was incubated for 1 h at 37 °C. Absorbance at 490 nm was assessed using a spectrophotometer SpectraCount (Packard Instrument, Meriden, CT, USA). Four technical replicates and three biological replicates were carried out, and the mean of these results was considered.

### 4.8. Cells Apoptosis Assay

In order to show the rate of apoptotic cells, 150,000 FMCm cells were cultured in a six-well plate. The cells were treated with HFn(DOX), DOX alone, and RPMI at the concentrations found to be significant in the proliferation assay: 48 h after treatment with 0.1 µM and 72 h after treatment with 0.01 µM and 0.1 µM. An Annexin V–Fluorescein isothiocyanate (FITC) Apoptosis Detection Kit (eBioscience, Thermo Scientific) was used on FMCm following the manufacturer’s instructions. Briefly, 200 µL of the cell suspension was centrifuged at 1100 rpm for 10 min at 4 °C, and this supernatant was discarded; after this step, 200 µL of binding buffer with 5 µL of annexin V–FITC was added, and the cells were incubated for 10 min at RT in the dark. After the incubation time, another 200 µL of binding buffer was added to the cell suspension that was subsequently centrifuged at 1100 rpm for 10 min at 4 °C, and the supernatant was discarded. After this step, 900 µL of binding buffer was added; then, the cells were acquired using a flow cytometer CyFlow Space (Partec-System, Sysmex Europe GmbH, Norderstedt-Amburgo, Germany) and the data were analyzed with the open-source software FCSalyzer (version 0.9.16-alpha). For each tube, 20,000 events were analyzed.

### 4.9. Statistical Analysis

Statistical analysis was performed using Prism 8.0.1 (GraphPad Software). To verify differences among groups, one-way ANOVA with Tukey’s multiple comparison test was used when values were normally distributed. Alternatively, the Kruskal–Wallis test with Dunn’s multiple comparison test was used. Normality distribution was established using a Shapiro test. The level of significance was fixed as *p* < 0.05.

## 5. Conclusions

Despite the needed additional studies, in the present work, the overexpression of TFR-1 was highlighted for the first time in feline mammary carcinoma, and the application of drug-loaded HFn nanoparticles was identified as a potentially useful treatment for this common malignancy in this animal species. Therefore, our results support an effort to explore the great potential offered by bionanotechnology for the design and development of innovative therapeutic approaches that could exhibit enhanced efficacy and target selectivity suitable to be evaluated in veterinary oncology.

## Figures and Tables

**Figure 1 cancers-13-01248-f001:**
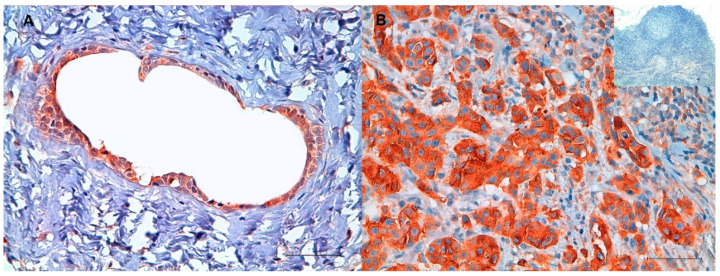
Immunohistochemical staining of transferrin receptor 1 (TFR-1). Immunohistochemistry was carried out on feline mammary gland. (**A**) Representative staining of healthy feline mammary gland tissue showing weakly positive cells (cytoplasmic immunolabeling). (**B**) Representative staining of a feline mammary carcinoma with lymph node metastasis showing strong cytoplasmatic immunolabeling of neoplastic cells associated with mild positivity of endothelial and inflammatory cells, as well as fibroblasts. The inset represents a feline lymph node, used as a negative control (10×). The images were obtained using a 40× objective. Immunohistochemistry TFR-1, Diaminobenzidine (DAB) chromogen. Scale bar: 50 µM.

**Figure 2 cancers-13-01248-f002:**
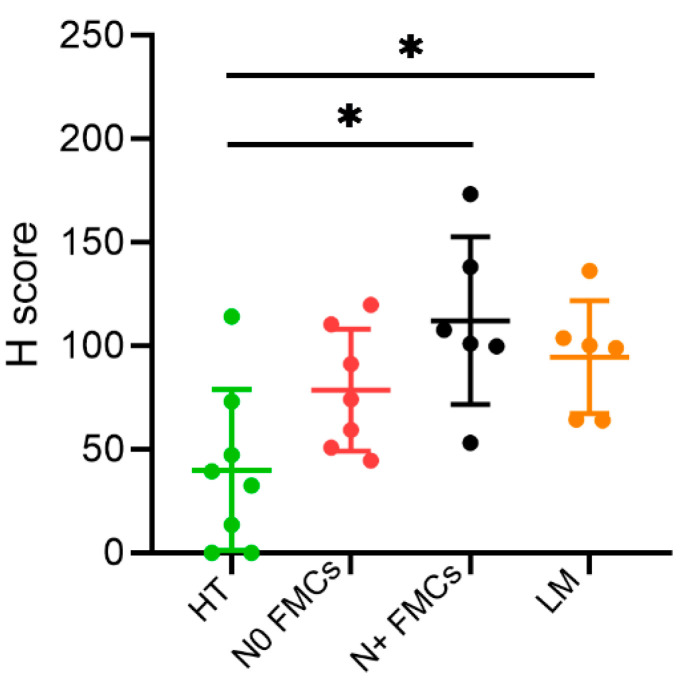
H-score of TFR-1 protein expression. H-score value of 27 samples of feline mammary gland tissue: healthy tissue (HT) (*n* = 8, green) (mean 40.07 ± SD 38.95), feline mammary carcinoma without lymph node metastasis (N0 FMCs) (*n* = 7, red) (mean 78.72 ± SD 29.41), feline mammary carcinoma with lymph node metastasis (N+ FMCs) (*n* = 6, black) (mean 112.28 ± SD 40.51), and lymph node metastasis (LM) (*n* = 6, orange) (mean 99.48 ± SD 27.95). The H-score value was significantly higher in N+ FMCs and in LM as compared to HT (* *p* < 0.05).

**Figure 3 cancers-13-01248-f003:**
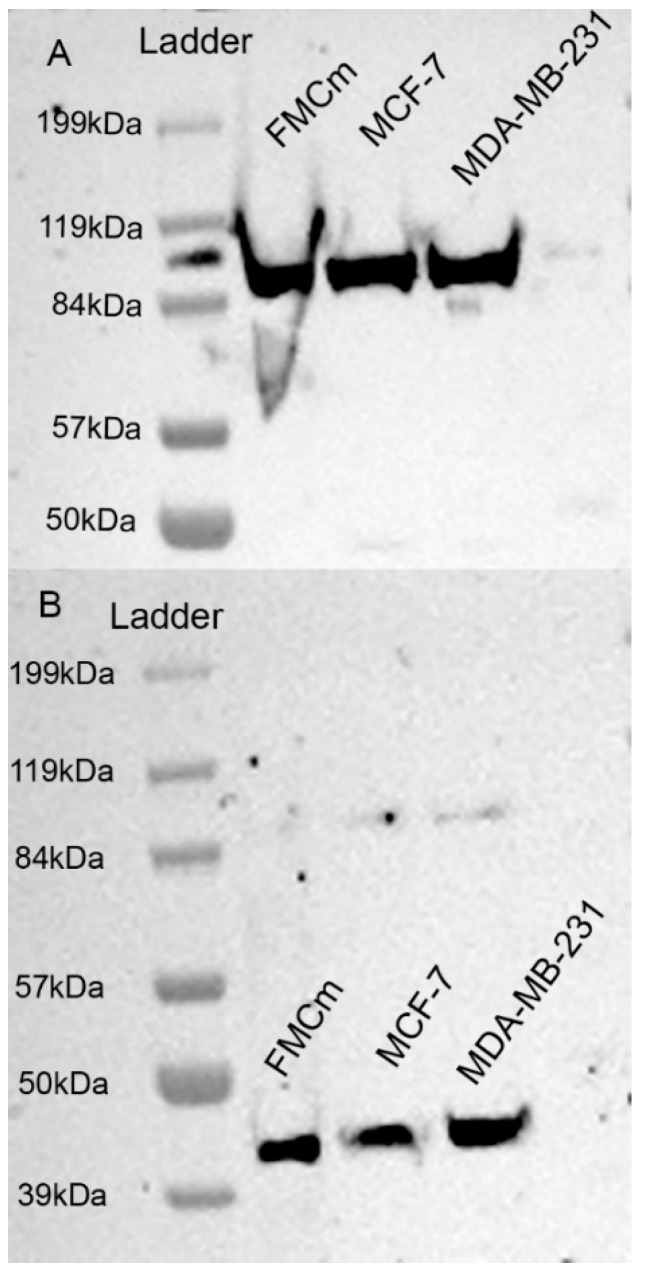
Western blotting analysis using antibody against TFR-1. (**A**) Western blotting analyses for cross-reactivity of the mouse anti-human TFR-1 antibody on feline proteins extracted from FMCm and human breast cancer cell lines, MCF-7 and MDA-MB-231. TFR-1 is shown as a 90 kDa protein. (**B**) Western blotting analyses for the housekeeping gene (beta-actin) on proteins extracted from FMCm, MCF-7, and MDA-MB231 cell lines. Beta-actin is shown as a 43 kDa protein.

**Figure 4 cancers-13-01248-f004:**
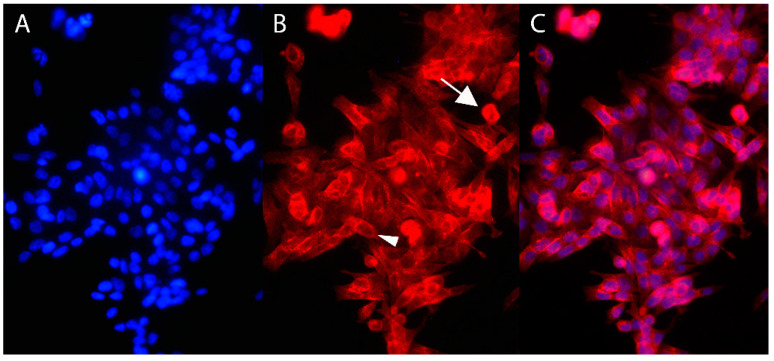
Immunofluorescence for TFR-1. Immunofluorescence carried out on FMCm (20×). (**A**) Diamidino-2-phenylindole (DAPI) immunostaining. (**B**) Cells were TFR-1^+^, with both cytoplasmic (arrow) and membrane localization (arrowhead). (**C**) Merged images.

**Figure 5 cancers-13-01248-f005:**
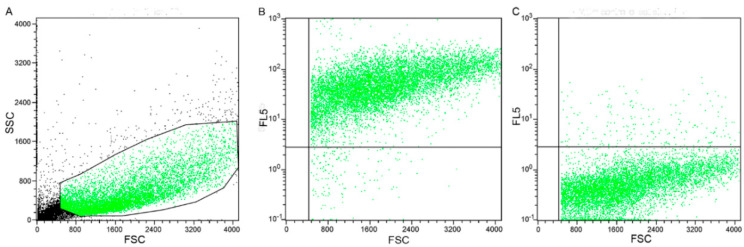
Flow cytometry for TFR-1. Flow cytometry performed on FMCm. (**A**) Morphological scatter plot (forward scatter, FSC; side scatter, SSC), showing cells with a cloud-shaped distribution; (**B**) scatter plot of FSC against FL5, showing cells positive for TFR-1; (**C**) scatter plot of FSC against FL5, i.e., the negative control, showing cells negative for TFR-1.

**Figure 6 cancers-13-01248-f006:**
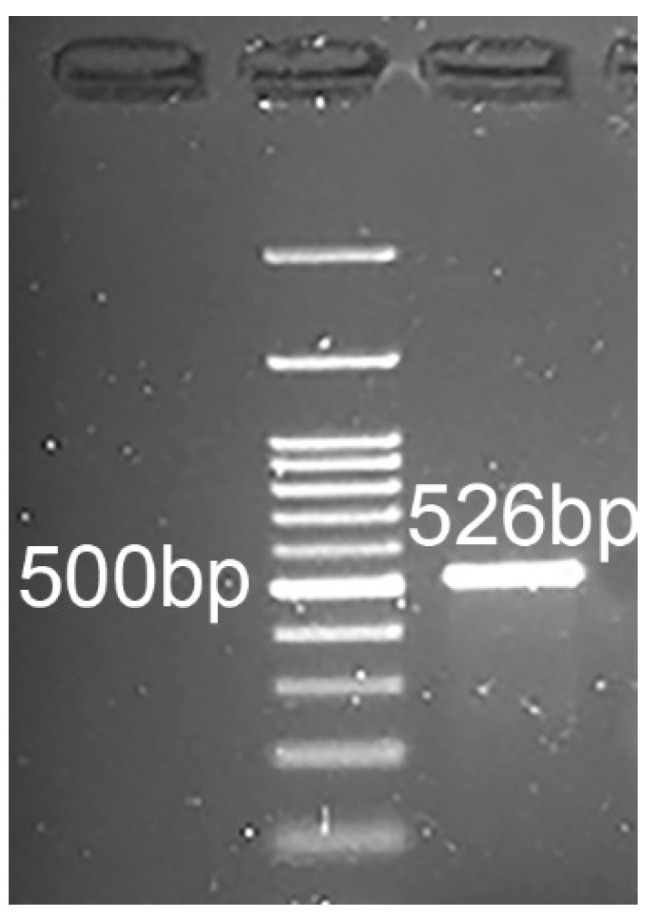
Genomic amplification of the *Felis catus* transferrin receptor gene (*TFRC*). RT-PCR analysis for *TFRC*. The amplified fragment size was 526 bp, as expected.

**Figure 7 cancers-13-01248-f007:**
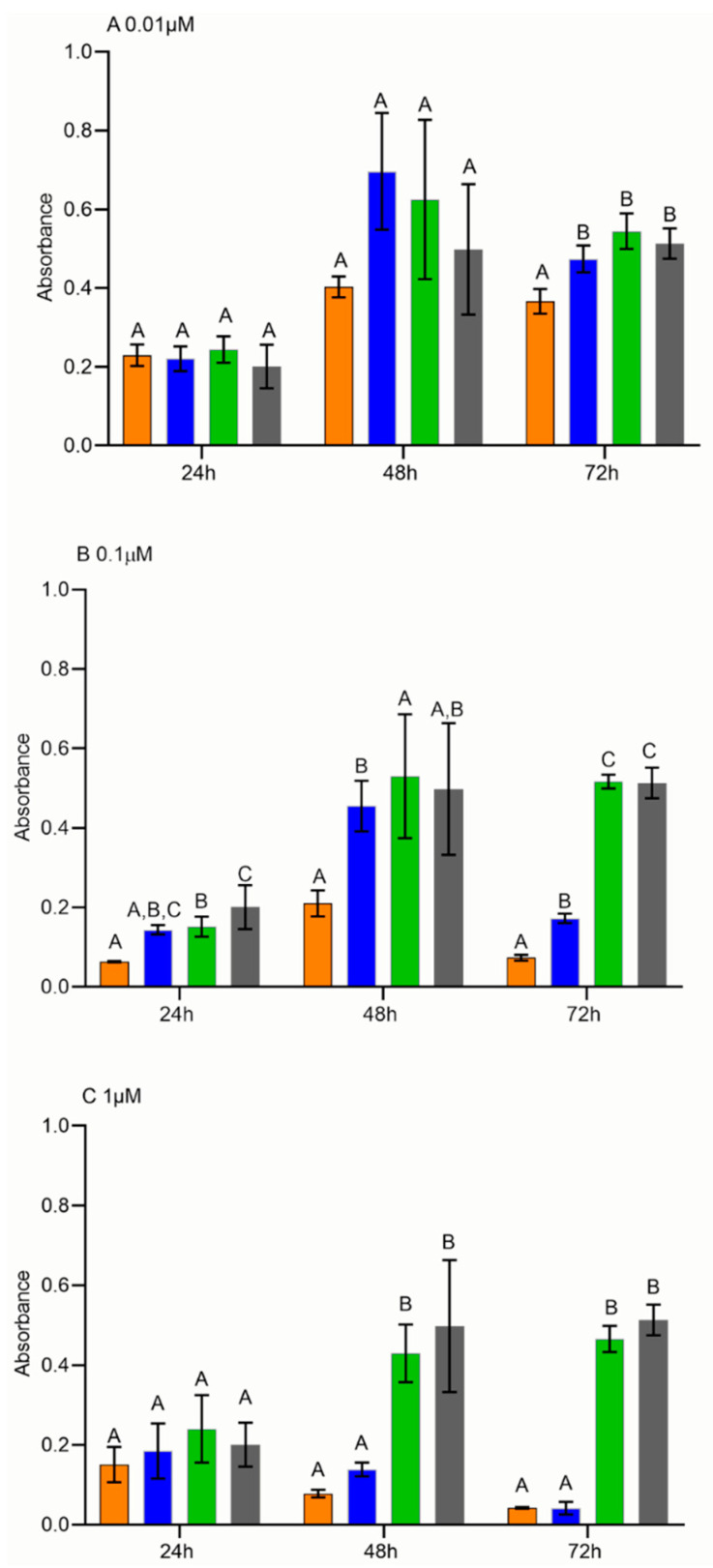
Cell proliferation assay. Proliferation assay on FMCm treated with engineered ferritin nanocage loaded with doxorubicin (HFn(DOX)) (orange), DOX alone (blue), HFn alone (green), and Roswell Park Memorial Institute (RPMI) (gray), at three different concentrations and three different time points: (**A**) proliferation assay at 0.01 µM; (**B**) proliferation assay at 0.1 µM; (**C**) proliferation assay at 1 µM. Cells treated with HFn(DOX) proliferated less than cells treated with DOX alone at 0.01 µM at 72 h post treatment (*p* < 0.05) (**A**) and at 0.1 µM after both 72 h and 48 h (B) (*p* < 0.05).

**Figure 8 cancers-13-01248-f008:**
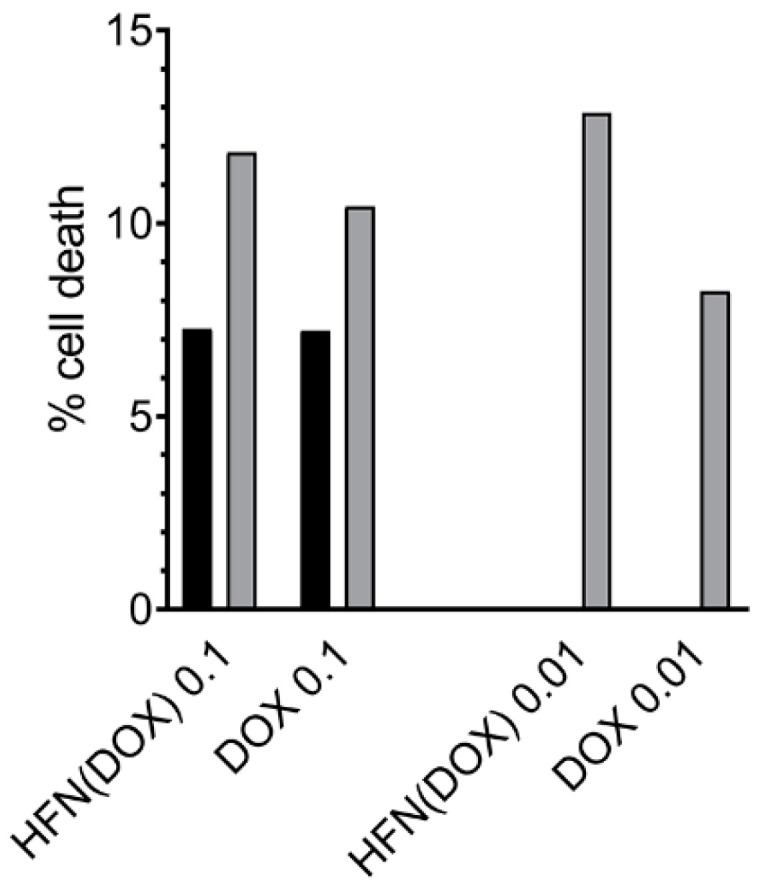
Cell apoptosis assay. Apoptosis assay on FMCm treated with HFn(DOX) or DOX alone at concentrations of 0.1 µM and 0.01 µM for 48 h (black) and 72 h (gray). HFn(DOX) caused more apoptosis than DOX alone 72 h after treatment at 0.1 and 0.01 µM.

## Data Availability

Data are contained within the article or Appendix A.

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
