# Peer review of "Evaluation of TFR-1 Expression in Feline Mammary Cancer and In Vitro Antitumor Efficacy Study of Doxorubicin-Loaded H-Ferritin Nanocages"

_cancers, 2021, doi:10.3390/cancers13061248_

Round 1

Reviewer 1 Report

Thank you for the opportunity to review this revised manuscript.

The authors answered satisfactorily to all the questions and remarks. However, the following points should be improved before the publication of the work:
1) the fact that FMC shows the same molecular subtypes as human breast cancer should be highlighted in the Introduction section (suggestions for reference: https://doi.org/10.1007/s13277-015-4251-z)
2) also that FMC has several biomarkers that can be used for diagnosis (https://doi.org/10.1038/s41598-020-60860-3). 

Author Response

We are very grateful for the review provided of this manuscript.

I added your useful suggestions into the introduction of the manuscript. Precisely, I changed the line 50, adding the information about the similarities between human breast cancer and feline mammary carcinoma.

FMCs share some characteristics with human breast cancer including molecular subtype and several biomarkers, used for diagnosis”

I really appreciate that you enjoyed the manuscript and thank you again for reviewing it.

Reviewer 2 Report

Compared to the submitted manuscript, the authors have conducted supplementary experiments in the revised manuscript, including the use of control breast cancer cell lines (MCF-7, MDA-MB-231), the measurement of apoptotic cell death under Doxorubicin or Doxorubicin/ H-ferritin treatment, and the addition of negative controls in the IHC figures. All of the modifications made by the authors are fully satisfactory in my opinion. This revised manuscript is original, very well designed and written, very informative. I strongly support publication of this revised manuscript in Cancers.

Remaining typos:

  • Page 9, line 231: “MDA-MB-231”
  • Page 13, line 414: “eBioscience”
  • Page 13, line 415: “manufacturer”

Author Response

We wish to thank you very much the #Reviewer 2.

We really appreciated that you had decided to support the publication of the article.

Thank you for your valuable and accurate suggestions. We really apologize for the typing errors. We corrected them and amended the manuscript accordingly.

Reviewer 3 Report

The study has interesting subjects and important to found pharmacological targets to mammary cancer in animal and Humans.

The authors improved this version, changed images, but we still have questions unsolved.

Figure 1 was completely reformulated, with better image quality, but appears to be manipulated, as the colour of the DAB is brown (as the older version), and this is orange.

The figure of the normal mammary gland displays some alterations, as duct ectasia. So, the question previously formulated about the normal mammary gland needs to be answered, as this image is the reflex of this absence of information.

The authors had improved the description of the material and methods section, but some questions are still without answer:

Were the mammary gland tumours selected all from different animals?

Was the normal mammary gland from healthy animals, or normal mammary gland near the tumour? And is it possible to know the oestrus cycle phase of this mammary tissue?

In immunohistochemistry technique is mandatory to refer to the kind of antigen retrieval made, what was the positive control and the counterstaining used.

In the discussion, the authors wrote:” In our case, no specific association was found between tumor histological grade, size or lymphovascular invasion (data not included) possibly because of the limited number of samples; still, there was an association with the metastatic tumor progression indicative of a more malignant behaviour”. But the authors didn´t describe the tumor histological grade, size or lymphovascular invasion in the material and methods, and this is a major point to discuss the results obtained..

Also, in the new information in the discussion, the affirmation “It could be assumed that cells treated with HFn(DOX) are less alive than cells treated with DOX alone”, is not clear. What is the meaning of “cells less alive”?

In the sentence: “calculating the rate of necrosis after 48h and 72h could not be reliable since a process defined as “late apoptosis/secondary necrosis” has been described to occur in vitro”, what is secondary necrosis? How can the authors confirm that this phenomenon as necrosis and not another kind of cell death, like necroptosis?

Author Response

We wish to thank you the #Reviewer 3 for reading and commenting the manuscript. Please see below the answer to your comments.

Figure 1 was completely reformulated, with better image quality, but appears to be manipulated, as the colour of the DAB is brown (as the older version), and this is orange.

Thank to your comment we can now better describe the uploaded images. As the previous images were overexposed and the background was not so clean, we decided then to run again the TFR-1 immunohistochemistry both on healthy mammary gland and on tumor. For this reason, the DAB stain is a bit different in the color tone. Also, in order to obtain a better image (background and contrast), parameters were adjusted in the image software (Leica DMD108) used to take the photos.

We hope that you appreciate the decision to make again the staining for having an image of better quality with no background discoloration.

The figure of the normal mammary gland displays some alterations, as duct ectasia. So, the question previously formulated about the normal mammary gland needs to be answered, as this image is the reflex of this absence of information.

We are not sure we understood this question. Sorry about this. The normal gland is mammary tissue of the same subject taken far from the tumor. It can display some non-neoplastic changes such as indeed duct ectasia for example due to some compression, usually in the cat it doesn’t display hyperplasia (as it is for the dog). Therefore, the non-neoplastic tissue is usually composed mainly by small ducts with lobules barely visible. In the previous figure 1A we chose a small lobule but this was no longer available into the section for the new staining. The previous figure 1A was a bit overexposed and so we could not manipulate that.

We hope this answer and contribute to your comment.

Were the mammary gland tumours selected all from different animals?

Thanks for the observation, however, the same question has already been formulated in the prior revision. We report here the answer:

The selection of the neoplastic tissues was made on different subjects. The patients underwent surgery for nodulectomy or monolateral or bilateral mastectomy, with ancillary lymph node excision as part of the diagnostic and treatment procedure. The tissues were submitted for histological examination and after diagnosis selected for the study. Unfortunately, feline mammary tumors are frequently very aggressive and therefore it was not possible to find a high number of low-grade tumors with no metastasis”.

Was the normal mammary gland from healthy animals, or normal mammary gland near the tumour? And is it possible to know the oestrus cycle phase of this mammary tissue?

Thanks to the reviewer 3 for pointing out this aspect, since this request has already been posed in the previous revision, we reported below the answer.

The healthy specimens included for the immunohistochemical analysis were obtained from the same animals that were submitted for surgical excision of the neoplastic masses. The non-tumoral mammary gland was peripheral to the tumor lesion. In cats, nodules can become very large and usually partial mastectomy is performed and normal mammary tissue far away from the lesion is included. Considering the oestrus cycle, since queens have induced ovulation, data on the oestral cycle phase should be reported by the owner due to behavioral changes, but this is an information which is indeed very rarely detected/reported”.

We hope that this comment can be accepted.

In immunohistochemistry technique is mandatory to refer to the kind of antigen retrieval made, what was the positive control and the counterstaining used.

In order to better specify the immunohistochemistry assay, here we report the answer to the previous reviewer, where we described better the protocol used:

Specifically, the staining procedure was performed using a semiautomatic immunostainer (Ventana Benchmark XT, Roche-Diagnostic). All reagents were dispensed automatically except for the primary antibody which was manually added. A kit containing the secondary antibody and a horseradish peroxidase (HRP) - conjugated polymer which binds mouse primary antibody (UltraView Universal DAB, Ventana Medical Systems) was used. The counterstaining with haematoxilin was automatically performed. Because in literature canine placenta is considered as a positive control for TFR-1[3], we used this section as a positive control too”.

Reference:

  1. Priest, H.; McDonough, S.; Erb, H.; Daddona, J.; Stokol, T. Transferrin receptor expression in canine lymphoma. Vet. Pathol. 2011, 48, 466–474, doi:10.1177/0300985810377074

In the discussion, the authors wrote:” In our case, no specific association was found between tumor histological grade, size or lymphovascular invasion (data not included) possibly because of the limited number of samples; still, there was an association with the metastatic tumor progression indicative of a more malignant behaviour”. But the authors didn´t describe the tumor histological grade, size or lymphovascular invasion in the material and methods, and this is a major point to discuss the results obtained.

Also, in the new information in the discussion, the affirmation “It could be assumed that cells treated with HFn(DOX) are less alive than cells treated with DOX alone”, is not clear. What is the meaning of “cells less alive”?

As reported in the text our results showed only a statistical significance when comparing the tumors with positive lymph nodes and healthy tissues and the lymph node mets against the healthy tissues. We analyzed correlation and differences between TFR-1 expression and the other parameters such as histological grade, size of the tumor and lymph vascular invasion but no statistical significance was found at all. For this reason we did not consider necessary to include data, since we did not want to comment too much on this result which might be biased by the number of samples. Regarding methodology the histological grading was performed according to the grading system of Peña et al., Vet Pathol. 2013 50(1):94-105 , the size was measured as the major diameter at histology and the lymph vascular invasion was observed around the tumor by optical microscopy. We include here our data, which were not required as supplementary material either.

To reply to the second comment instead, the term “less alive” referred to the fact that the cells treated with HFn(DOX) manifested lower absorbance in the performed assay and this indicate less metabolic activity which is considered proportional by this assay with the proliferation activity. We are aware that this might be slightly imprecise but we wanted to summarize less metabolism/less proliferation as generally being less alive. To verify that the lower proliferation/metabolism of cells treated with HFn, was linked to a greater cell death, compared to cells treated with the drug alone, as previously requested by a reviewer, we evaluated the rate of apoptosis. This data confirmed that nanocage loaded with doxorubicin causes more cell death than cells treated with only DOXO.

In the line 42, we wrote: “the use of HFn(DOX) resulted in less proliferation of cells and increased apoptosis when compared to the drug alone”

In the sentence: “calculating the rate of necrosis after 48h and 72h could not be reliable since a process defined as “late apoptosis/secondary necrosis” has been described to occur in vitro”, what is secondary necrosis? How can the authors confirm that this phenomenon as necrosis and not another kind of cell death, like necroptosis?

The rate of cell death has been evaluated by Annexin V-FITC Apoptosis Detection Kit (eBioscinece). The Annexin V binds phosphatidylserine that during apoptosis traslocate in the external layer of the cellular membrane.

I reported here the protocol of this kit, wrote in the material and methods section of the manuscript; line 412.

“Briefly, 200 µl of cell suspension were centrifuged at 1100 rpm for 10 minutes at 4°C and supernatant was discarded, after this step 200 µl of binding buffer with 5 µl of annexin V-FITC were added, and the cells were incubated for 10 minutes at RT in the dark. After the incubation time, another 200 µl of binding buffer were added to the cell suspension that was subsequently centrifuged at 1100 rpm for 10 minutes at 4°C and the supernatant was discarded. After this step, 900 µl of binding buffer were added and, then the cells were acquired by flow cytometer CyFlow Space (Partec-System, Sysmex Europe GmbH, Norderstedt-Amburgo, Germany) and the data were analyzed with the open-source software FCSalyzer (version 0.9.16-alpha). For each tube, 20,000 events were analyzed”.

It is has known that there are several types of cell death as apoptosis, necrosis and more recently was found also necroptosis. The apoptosis is characterized by formation of apoptotic bodies and activation of caspases. Necrosis instead, is unregulated mechanism with disruption of cellular membrane. Necroptosis, is a regulated cell death because is tumor necrosis factor induced [1]. Recently, it was found that in cell cultures after a time interval such as 48 and 72 hours there is a phenomenon called late apoptosis/secondary necrosis. This process produces membrane disruption of apoptotic bodies that in cell culture conditions cannot be removed by phagocytic cells [2]. We relayed on the literature to mention this possible process that would alter the evaluation of necrosis. Further additional more detailed studies should then be performed to evaluated precisely necrosis and necroptosis rates in our cell cultures. Since doxo with and without the nanocage has been described as acting mainly via apoptosis [3] we focused on this process.

We hope that this explanation is enough to answer your question.

References:

  1. Koopman, G.; Reutelingsperger, C.P.M.; Kuijten, G.A.M.; Keehnen, R.M.J.; Pals, S.T.; Van Oers, M.H.J. Annexin V for flow cytometric detection of phosphatidylserine expression on B cells undergoing apoptosis. Blood 1994, 84, 1415–1420, doi:10.1182/blood.v84.5.1415.1415.
  2. Berghe, T. Vanden; Vanlangenakker, N.; Parthoens, E.; Deckers, W.; Devos, M.; Festjens, N.; Guerin, C.J.; Brunk, U.T.; Declercq, W.; Vandenabeele, P. Necroptosis, necrosis and secondary necrosis converge on similar cellular disintegration features. Cell Death Differ. 2010, 17, 922–930, doi:10.1038/cdd.2009.184.
  3. Bellini, M.; Mazzucchelli, S.; Galbiati, E.; Sommaruga, S.; Fiandra, L.; Truffi, M.; Rizzuto, M.A.; Colombo, M.; Tortora, P.; Corsi, F.; et al. Protein nanocages for self-triggered nuclear delivery of DNA-targeted chemotherapeutics in Cancer Cells. J. Control. Release 2014, 196, 184–196, doi:10.1016/j.jconrel.2014.10.002.

This manuscript is a resubmission of an earlier submission. The following is a list of the peer review reports and author responses from that submission.

Round 1

Reviewer 1 Report

The manuscript submitted by Rensi et al. entitled "Evaluation of TFR-1 expression in feline mammary cancer and in vitro antitumor efficacy study of doxorubicin-loaded H-Ferritin nanocages" aims to demonstrate that DOX-loaded in H-Ferritin nanocages has major efficacy that DOX alone and to reveal the expression of TFR-1 in feline mammary cancers.

In the reviewer's opinion, the MS and the study has major flaws that can not be suppressed:

1 - The work doesn't have any new hypothesis. All the results reported were described in humans;

2 - The antibody used against TFR-1 is not validated to feline tissue samples;

3 - The in vitro study was performed using only one cell line!

4 - Controls are missing (e.g. normal mammary feline cell line);

5 - The methodology used is not soundness; 

6 - Several references are missing, regarding the feline mammary carcinoma;

Reviewer 2 Report

The authors report first on transferrin receptor-1 (TFR-1) expression in feline mammary carcinomas (FMCs) compared to the normal mammary gland, which has been investigated only once previously, without significant results. Here, the authors demonstrate that TFR-1 is more expressed in node-positive FMCs and FMC nodal metastases than in the normal mammary gland.

The authors then report on TFR-1 expression in a metastatic feline mammary carcinoma cell line (FMCm), and take advantage of TFR-1 expression to use a doxorubicin/H-ferritin drug conjugate (HFn(DOX)), compared to doxorubicin alone (DOX), as a cytostatic drug.

The manuscript is very well written, the authors have demonstrated the high degree of homology between human and feline TFR-1 proteins and genes, and the results are clearly presented. Although I have some major reservations on the authors’ interpretation of some of the results presented, I think that this study is very informative, and innovative in the field of veterinary mammary oncology.

Major comments:

  1. The authors’ interpretation “our results highlighted an increased TFR1 expression associated to an increase in tumor malignancy” (lines 39-40) is not completely supported by the results presented. “Tumor malignancy” evokes the histological grade of malignancy, which is not described in this manuscript. The authors have demonstrated instead that the level of TFR1 expression by IHC increased with advancing stages (defined by the presence/absence of lymph node metastasis). Moreover, there were no attempts at associating expression levels of TFR1 with the histological grade, tumor size, lymphovascular invasion, ER, PR, or HER2 expression in this study.
  2. The authors’ conclusion “the use of HFn(DOX) caused drug internalization in cancer cells more efficiently/rapidly than administering the drug alone” is not supported by the results presented. Doxorubicin internalization was not investigated in this study. Page 6, lines 158-165, cell proliferation assay: the authors have chosen cell proliferation assays in order to compare DOX and HFn(DOX) efficiency on FMCm cells. This is fine, but not completely informative. Measurement of apoptotic and necrotic cell death could have been considered, as well as determination of the LD50 values for DOX and HFn(DOX). Comparisons between 2 cytotoxic compounds without determination of cell mortality and LD50 values represent an overinterpretation of the results presented.

Minor comments:

  1. Throughout the text: the use of “primary metastatic” terminology for FMCs is confusing compared to human breast cancers, where “primary” and “metastatic” designate different tumor locations, and where “metastatic” designates the presence of distant metastases, not nodal metastases. Maybe consider the use of “FMCs with nodal metastasis” or “N+ FMCs” for “primary metastatic FMCs”, and “FMCs without lymph node metastasis”, or “N0 FMCs”, for “non-metastatic primary FMCs”.
  2. Pages 9-10, lines 268-281, immunohistochemistry: what was the negative control? Did the authors replace the primary antibody by an isotype-matched irrelevant immunoglobulin used at the same concentration?
  3. Page 3, Figure 1: Figure 1A is overexposed. There seems to be also a (non-specific?) staining of the nuclei of luminal cells, can the authors confirm and discuss this aspect? Was TFR1 expression in fibroblasts and endothelial cells expected? Is there a possibility that some of the staining observed is non-specific? Maybe add an inset with the negative control in order to show that the TFR1 signal is specific.
  4. Page 3, Figure 1, and page 5, Figure 4: if possible, please add scale bars to these Figures.
  5. Page 3, results, immunohistochemistry: can the authors specify, either in the text of in the legend for Figure 2, the mean ± SD values of the H-scores for TFR1 expression?
  6. Page 4, lines 26-127, Western Blot: can the authors replace the reference of the primary antibody by a reference indicating that 90 kDa is the expected molecular weight of feline TFR1?
  7. Page 4, Figure 3, Western Blot: I think that a known positive control should have been used together with the FMCm cell line, for instance a breast cancer cell line with known TFR1 overexpression. Moreover, actin was not used as a loading control for the Western Blot, and thus the intensity of the 90 kDa band cannot be compared to a housekeeping protein.
  8. Page 5, results, flow cytometry (lines 140-144) and pages 10-11, methods, flow cytometry (lines 312-329): in the methods, the negative control is described as omission of the secondary antibody, while in the results the use of an isotype control is mentioned: please harmonize. For flow cytometry, the use of a positive control (a breast cancer cell line with TFR1 overexpression) would have been interesting, to indicate whether TFR1 levels in FMCs are relatively low or relatively high.
  9. Missing in the discussion: while the authors state “TFR-1 overexpression in malignant tumors has been already exploited in human medicine to internalize anticancer drugs more selectively and at a higher dosage in tumor cells” (lines 199-200), TFR1 overexpression was not demonstrated in FMCs of the present study. Firstly, because the level of TFR1 expression was not significantly different in healthy tissue and in node-negative FMCs (Figure 2). Secondly, because the H-score obtained in the present study (maximum around 170 points according to Figure 2) are far from the scores obtained with protein overexpression. Is it possible then to discuss (1) in breast cancers, is there an association between TFR1 expression by cancer cells and the effectiveness of HFn(DOX)? and (2) whether or not the TFR1 expression levels found in FMCs would be sufficient to allow for HFn(DOX) internalization? How many of the investigated FMCs can be considered TFR1-overexpssing FMCs?

Typos

  • Page 9, line 255: “spares” (with s).
  • Page 10, line 273: “UltraView” (without s).
  • Page 10, line 305: a parenthesis is missing after “RT”.
  • Page 11, line 323: the dot after “negative control” may be replaced by a comma.

Page 13, lines 464-465, reference 35: the

Reviewer 3 Report

The study has interesting subjects and important to found pharmacological targets to mammary cancer in animal and Humans.

The authors need to know that the techniques used must be described for the complete understanding of the readers. If the techniques were made based on consulted papers, the authors should reference them, but need to describe summarize the technique, in a way that the reader doesn’t need to go the original paper to understand what was made.

The authors had as biological material, 21 feline mammary cancer, and a feline cell line of mammary tumor.

They found and overexpression of transferrin receptor 1 (TR-1) on the mammary cancer tissues and its lymph node metastasis.

The authors employed several different techniques (as in vitro proliferation assays, Western Blotting analysis, Gene Expression and Sanger Sequencing, flow cytometry, Immunohistochemistry and Immunofluorescence) to prove the presence of this receptor, and also the overexpression of that on mammary tumors.

The study is well constructed and the paper is well written.

Nevertheless, some points need to be elucidated, and complete:

The tissues selected were all from different animals?

The normal mammary gland was from healthy animals? And is possible to know the oestrus cycle phase of this mammary tissue? How it was collected this normal gland?

Why the authors don’t have benign lesions? Why the groups only have only 6-7 lesions each?

The lymph node metastasis was from the same tumors used in this study as “carcinoma with metastasis”?

Why this study was made only with simple carcinomas?

In the description of the Immunohistochemistry technique, the authors miss some information: What kind of antigen retrieval was made? What was the incubation time of the primary antibody? What was used as a positive control? What was the counterstaining?

The H score obtained for evaluation needs more explanation:

            What as the magnification for the evaluation?

            What were the cells analysed for the evaluation to intensity and percentage? Only the epithelial cells? Stroma cells too?

            How was obtained the final score, between 0-300?

For the cell culture description is need to know if this cell culture was purchased or made by the authors. And the technique needs to have information about the number of cells/well or ml and time of cell culture.

For the Western Blotting analysis, how many cells were used? And the protocol needs to be, at least, summarized.

The same antibody was used in different techniques. Why the concentration were different?

For the cell proliferation assay, how many cells were used per well or ml? How many copies (wells) were made for every assay? (For every component, dilution, day…)

The results need to be improved with respect to gene expression of the TFR-1.

The histological images of figure 1 seem to have different magnification. And the normal gland doesn’t seem like a normal, resting, gland. What was the oestrus cycle stage of this animal?

The authors must remove the reference to the figures throughout the discussion.

What is the utility of the confocal microscope in this study and how it will be useful?